# Multiuser Incomplete Preference *K*-Nearest Neighbor Query Method Based on Differential Privacy in Road Network

**Liping Zhang [1], Xiaojing Zhang [1] and Song Li [1,***

School of Computer Science and Technology, Harbin University of Science and Technology, Harbin 150080, China;
zhangliping0730@hrbust.edu.cn (L.Z.); 2020410065@stu.hrbust.edu.cn (X.Z.)
* Correspondence: lisongbeifen@hrbust.edu.cn

**Abstract:** In view of the existing research in the field of *k*-nearest neighbor query in the road network, the incompleteness of the query user's preference for data objects and the privacy protection of the query results are not considered, this paper proposes a multiuser incomplete preference *k*-nearest neighbor query algorithm based on differential privacy in the road network. The algorithm is divided into four parts; the first part proposes a multiuser incomplete preference completion algorithm based on association rules. The algorithm firstly uses the frequent pattern tree proposed in this paper to mine frequent item sets, then uses frequent item sets to mine strong correlation rules, and finally completes multiuser incomplete preference based on strong correlation rules. The second part proposes attribute preference weight coefficient based on multiuser's different preferences and clusters users accordingly. The third part compares the dominance of the query object, filters the data with low dominance, and performs a *k*-neighbor query. The fourth part proposes a privacy budget allocation method based on differential privacy technology. The method uses the Laplace mechanism to add noise to the result release and balance the privacy and availability of data. Theoretical research and experimental analysis show that the proposed method can better deal with the multiuser incomplete preference *k*-nearest neighbor query and privacy protection problems in the road network.

**Keywords:** incomplete preference; spatiotemporal association rule; *k* nearest neighbor query in road network; privacy protection

## 1. Introduction

In recent years, the *k*-nearest neighbor (kNN) query is still an important research problem in the field of spatial query. With the deepening of research, the kNN query based on Euclidean space develops into a kNN query based on the road network and has many variants, such as high-dimensional approximate *k*-nearest neighbor query [1], continuous *k*-nearest neighbor query [2], line segment *k*-nearest neighbor query in obstacle space [3], probability threshold *k*-nearest neighbor query [4], etc. However, the above literature does not consider the user's preference in the *k*-nearest neighbor query, which cannot better meet the query requirements of users. Based on this, Yiu et al. [5] first studied the query problem of spatial preference, divided the objects into data objects and feature objects, and indexed them respectively. In order to improve the efficiency of the algorithm, Rocha-Junior et al. [6] proposed a spatial preference query method based on materialization technology, which significantly reduces CPU and I/O costs. Wang et al. [7] put forward a technique for processing authenticated user-defined query functions, which can process queries on data calculated by user-defined mathematical functions. However, these queries only focus on a single query user associated with a given location, ignoring the problem of a group of users' query and their preferences. For example, multiple users want to query multiple milk tea shops close to their current location; due to the uncertainty of the individual user's preference for "cheap price," this preference is missing, so in addition to "close distance" and "milk tea shop" for the user's preference, "cheap price" is also likely to be one of



his preferences. Therefore, how to mine the potentially uncertain personal preferences of multiple users to query multiple interest points (data points) that meet multiple users has become a noteworthy problem. Therefore, the algorithm proposed in this paper can be applied to this background. The main contributions of this paper are as follows:

(1) Aiming at the problem of the incompleteness of query users' preference, this paper proposes a completion algorithm for multiuser incomplete preference, which proposes the concept of frequent pattern tree HUFP-tree and mines frequent itemsets and strong association rules based on this. Then it completes the incomplete data of multiple users according to the association rules, which reduces the complexity of the query and improves the accuracy of query results.

(2) Aiming at the difference in the user's preference for multiple queries, this paper proposes a clustering and grouping algorithm based on multiuser preference. According to the proposed concept of multiuser preference weight coefficient, the algorithm classifies the attributes of data objects by multi-query users and then groups them within the cluster for the road network distance between query users, which improves the accuracy and efficiency of query results.

(3) In order to improve the efficiency of the multiuser incomplete preference $k$ neighbor query algorithm in the road network environment, this paper proposes a reduction and refining algorithm for data objects, which first prunes the data objects according to the distance of the road network, then deletes the data points with weak dominance, which can filter out a large number of unqualified data objects, and finally performs a $k$ neighbor query on the data objects after refining them to obtain the query results. This method effectively reduces the query time and improves the query efficiency.

(4) In order to protect the privacy of the query result data in the $k$-nearest neighbor query with an incomplete preference of multiple users under the road network environment, this paper proposes a privacy budget allocation algorithm. This uses differential privacy technology to add Laplace noise to the query results according to the weight ratio of the results, effectively protecting the privacy of the data.

Compared with the previous methods, the operation efficiency of the proposed method is improved by 15%. In terms of privacy protection, when $\varepsilon$ is between 0.1 and 1, there is better privacy. The rest of the paper is organized as follows. Section 3 of this document gives important definitions. Section 4.1 proposes a multiuser incomplete preference completion algorithm based on association rules. Section 4.2 proposes user clustering and intraclass user grouping algorithms based on attribute preference weight coefficients. Section 5.1 proposes a $k$-nearest neighbor query algorithm with an incomplete preference for multiple users in the road network. Section 5.2 provides privacy budget allocation and privacy and usability analysis. Section 6 gives the corresponding experimental analysis. Section 7 gives a summary. The method proposed in this paper is 15% more efficient than the previous method; in terms of privacy protection, there is better privacy when $\varepsilon$ is between 0.1 and 1. The specific process is shown in Figure 1.

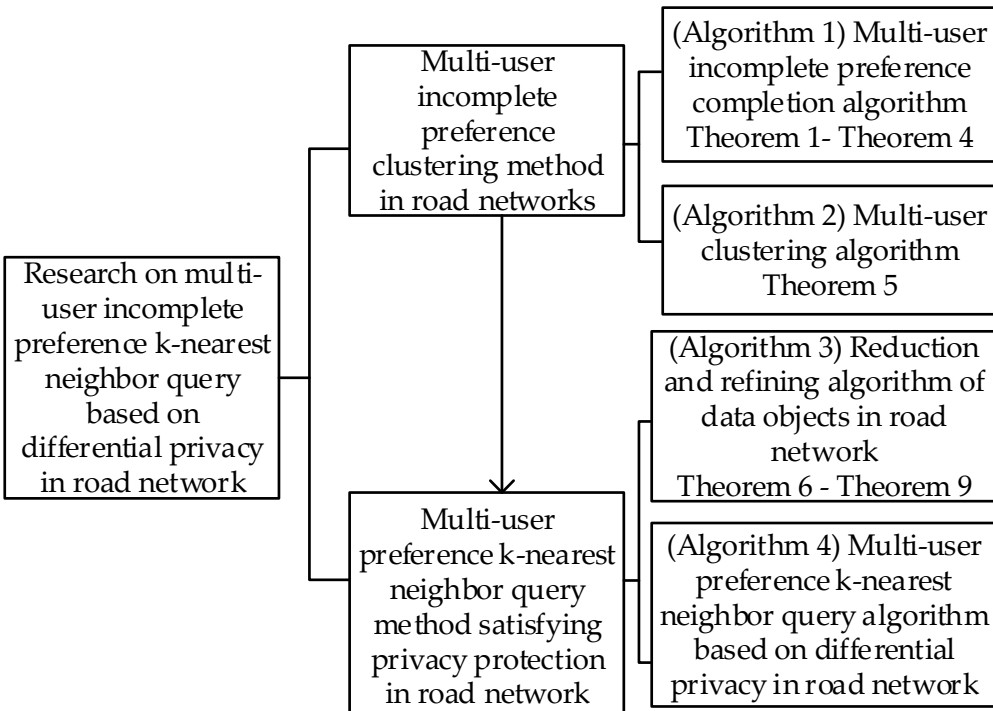

**Figure 1.** Algorithm relationship and data processing flow.

## 2. Related Work

*K*-nearest neighbor (*k*NN) query has always played a very important role. For example, with the widely used internet of things, 5G, and smart city technologies, in order to acquire a variety of vehicle trajectory data, researchers construct the *k*-nearest neighbor-based internet of vehicles in a dynamic manner to achieve the effect of vehicle trajectory clustering. Specifically, they first construct the *k*-nearest neighbor-based internet of vehicles in a dynamic manner. Then they learn the low-dimensional representations of the vehicles by performing dynamic network representation learning on the constructed network. Finally, using the learned vehicle vectors, vehicle trajectories are clustered with machine learning methods; the method has good performance, but its accuracy is low [8]. In order to solve the K-nearest neighbor query problem of time-dependent road network, Yang et al. [9] proposed the index based on Voronoi graph and V-tree index. They preprocessed the time-dependent road network, modeled the road network as a graph with time information, and proposed a k-nearest neighbor query algorithm. The algorithm calculates the minimum travel function between vertices in advance, according to which a dynamic index based on Voronoi graph is designed. Through this index, the position of a certain vertex can be determined at any given time point. This method improves the query efficiency, but the query accuracy is low, Li et al. [10] proposed a location-aware spatial preference query, which is the first query that combines group query and preference query. In this query, the spatial distance and POI category preference of all users in the group are considered, but the time complexity is high. Cai et al. [11] designed a spatial-social index to maintain users' social, spatial, and textual information and develop an effective solution based on the index to solve the problem of maximizing the overall impact. Because existing research methods have not completely solved the problem of variable clustered nearest neighbor query ($F_{ANNR}$) in a road network, Chen et al. [12] proposed a $kF_{ANNR}$ query based on keyword aware variant on this basis, but this query does not fully consider the preference of each user. In order to solve the problem of user community spatial preference and improve its performance, Wang et al. [13] proposed a spatially aware user community preference query algorithm based on a knowledge graph, which considers the user's location semantic information and POIs to effectively discover the user's community

preference. In order to achieve this goal, the algorithm first used Tr-tree spatial index to improve the query efficiency and then introduced the community satisfaction degree model based on the knowledge graph to comprehensively evaluate whether POIs can better meet the preference of the user community. Wang et al. [14] proposed an algorithm to find the optimal result of multiuser positional keyword queries based on the minimum cost function. Wang et al. [15] Saad N H M et al. proposed the SkyQUD algorithm, which divided the data set according to the characteristics of each query object, and in the pruning process, it used a probability threshold τ to limit the possibility of a certain dimension in the data set to become the user's preference, but the query accuracy of this method was low.

In view of the incompleteness of multiuser preference, O'connor et al. [16] put forward the group recommendation method, which provides suggestions for groups based on users' historical information. Amer-Yahia et al. [17] proposed a group recommendation algorithm, and Yu et al. [18] proposed a quantitative conditional preference model, which creates a "pseudo-user" profile for each group or merges the recommendation list of a single user at run time. However, group recommendation largely depends on the user's past preference, query history, and other relevant information. Zheng et al. [19] proposed a keyword query based on top-*k* location, but this method is not suitable for a group of users because two users in a group may choose different POIs that reflect personal preference. Therefore multiuser preference queries require a mechanism to balance the preference of all users in a group.

In order to protect the privacy of data, Dritsas et al. studied the problem of privacy-preserving on spatiotemporal databases, applied *k*-Nearest Neighbor (*k*-NN) algorithm on the whole dataset, investigated the *k*-anonymity of mobile users based on real trajectory data. The *k*-anonymity set consists of the *k*-nearest neighbors. They constructed a motion vector of the form (x, y, g, v) where x and y are the spatial coordinates, g is the angle direction, and v is the velocity of mobile users, and studied the problem in four-dimensional space. This method can highly preserve *k*-anonymity, but its accuracy is poor [20]. Further, Dritsas et al. proposed the DUST framework, by which a raw trajectory is split into a number of linear sub-trajectories, which are subjected to the dual transformation that formulates the representatives of each linear component of the initial trajectory, and in this way, the proposed approach is expected to reinforce the privacy protection of such data [21].

In [22], they aimed at the privacy protection problem in preference-based recommendation systems and designed an encrypted index based on B$^+$tree to protect the privacy of data sets. Through this encrypted index, a group *k*-nearest neighbor query (GkNN) based on user preference can be performed by filtering and verification, but the query does not consider the overall preference of each group. In addition, Zhou et al. [23] used the Paillier homomorphic encryption algorithm to ensure that each user's location and query content in the kNN query is protected from privacy attacks, but this method is vulnerable to attacks with a knowledge background. Furthermore, this solution is inefficient in terms of LSP computation cost because it requires a lot of hypothesis testing. Although these methods can ensure security, they often lead to high communication costs and computational complexity, which seriously affect big data analytics. Differential privacy [24] is a technology that can guarantee user privacy from arbitrary background knowledge attacks. In this context, many studies have proposed the privacy protection method of information theory [25] to quantitatively measure privacy leakage and design privacy protection mechanisms based on these measures. Although the concept of differential privacy is more rigorous than the information theory approach, it is more in line with the practical requirements of many application fields.

Therefore, based on the above considerations, Kaaniche et al. [26] designed a privacy protection framework based on user preference and proposed a secure computation based on collaboration to interfere with the configuration file. The computation includes intermediate nodes between the client and the service provider, but this method will generate more extra computation overhead. Rahali et al. [27] proposed a new recommendation

algorithm for privacy protection, which relies on local differential privacy (LDP), and the query noise preference profile is transmitted to the service provider as a disturbed Bloom filter, which is suitable for classification and clustering tasks under local differential privacy. Zhang et al. [28] improved the privacy budget allocation scheme based on the differential privacy clustering algorithm DPk-means. This method breaks down the total privacy budget into decreasing arithmetic progressions and improved usability of clustering results. Fan et al. [29] improved the privacy budget allocation scheme based on the differential privacy clustering algorithm DPk-means, decomposed the total privacy budget into decreasing arithmetic progression, and proposed a differential privacy clustering algorithm APDPk-means based on arithmetic progression privacy budget allocation, which improved the availability of clustering results. Zhou et al. [30] proposed the privacy protection framework of distributed recommender system; according to the federated learning strategy, this framework deploys a Laplace mechanism for each agent to train local history and introduces an adaptive binary tree noise aggregation method, which ensures differential privacy and also avoids performance loss.

At present, there are two studies on the uncertainty [31] and incompleteness of user preference. However, in geographic social networking applications, users' personal preferences are latent, and it is difficult for users to express them accurately. Since the inaccurate expression of individual user preference in multiusers will lead to inaccurate or even incorrect query results, how to mine the potential personal preference of group users has become a key problem.

In addition, when querying based on the preference of multiple users, not only the location of users will be subject to privacy attacks, but also the query content of users is more vulnerable to security threats when data is released. Therefore, it is increasingly important to protect user data privacy from attacks when publishing user data. Therefore, in order to fully explore the user's personal preference and protect the privacy of data in the *k*-neighbor query, this paper proposes the *k*-neighbor query method based on differential privacy based on the incomplete preference of multiple users in the road network.

## 3. Definitions and Symbol Descriptions

According to the content of the research and related technologies applied, this section provides the following basic definitions.

In the traditional *k*-nearest neighbor query, in order to find the nearest *k* interest points (data points) from the interrogator, many researchers treat the "distance" problem as the "distance" in Euclidean space to simplify the problem. However, this processing method does not consider the real distance between roads in practical application, so the applicability is poor. Therefore, the application background in this paper is the road network environment, indicating that the real distance between roads is adopted when querying the nearest *k* points of interest (data points) in the aspect of "distance." In addition, we use the "network Voronoi diagram" technology to better represent the road network.

**Definition 1.** *Network Voronoi Polygon (NVP). The NVP is generated from the data points in the road network, and the NVP forms the network Voronoi diagram. The NVP generated by data point $p_i$ is denoted as VNVP($p_i$). If the data set of the given road network is P = {$p_1$, $p_2$, . . ., $p_n$}, edge set is {$e_1$, $e_2$, . . ., $e_k$}, $p_j \neq p_i$, then VNVP($p_i$) can be formalized as:*

$$V_{\text{NVP}}(p_i) = \{p | p \in \bigcup_{i=1}^{k} e_i, d(p, p_i) \leq d(p, p_j)\} \tag{1}$$

All data points on the road network correspond to the NVP region $V_{\text{NVP}}(p) = \{V_{\text{NVP}}(p_1),$ . . ., $V_{\text{NVP}}(p_n)\}$ is the network Voronoi diagram, as shown in Figure 2.

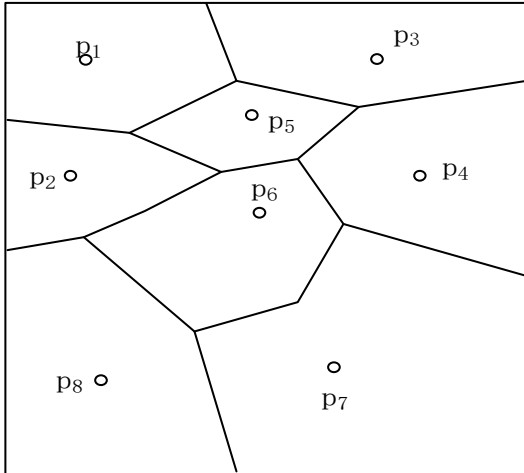

**Figure 2.** Network Voronoi diagram.

The network Voronoi diagram has the following properties [32]:

(1)   Each NVP corresponds to only one data point.
(2)   The edge of the Voronoi network graph is formed by the vertical bisector of two adjacent data points. The distance between the data points at both ends of the vertical bisector and the road network of the vertical bisector is equal.
(3)   The network Voronoi diagram divides the space into non-overlapping NVPS and the road network distance from any point in each NVP to the NVP generation point is smaller than the distance to the other NVP generation points.

**Definition 2.** *K-nearest neighbor query based on multiuser incomplete preference. Given a collection of query users $U = \{u_1, u_2, \ldots, u_i\}$, data object $O = \{o_1, o_2, \ldots, o_i\}$, $o_i = \{o_i.w_1, o_i.w_2, \ldots, o_i.w_j\}$, $u_i = \{<u_i.id>, <u_i.w_1, u_i.w_2, \ldots, u_i.w_l>\}$, $u_i.id$ is the unique identifier of the query user, $u_i.w_l$ is a query user's query preference for data object attributes, and the k-nearest neighbor query returns a set R of k data objects. None of the data objects in R will be dominated by other data objects in O, i.e., $R = \{o_i | \{o_j \in O, o_j \preceq o_i\}\}$ and $u_i.w_l \in o_i.w_j$.*

**Definition 3**. *Degree of Support and Confidence [1]. The degree of support is the probability that the rule $X \rightarrow Y$ in the transaction set T contains both the itemsets X and Y, denoted as $sup(x \rightarrow y)$, such that*

$$sup(x \rightarrow y) = \frac{|x \cup y|}{T} \qquad (2)$$

Confidence refers to the probability of including itemset *Y* if it contains itemset *X*, that is:

$$con(x \rightarrow y) = \frac{sup(x \rightarrow y)}{sup(x)} \qquad (3)$$

Support and confidence are two important concepts in association rules. This paper obtains association rules based on support and confidence to complete the incomplete preferences of multiple users.

Considering the privacy of incomplete data, this paper uses differential privacy technology to protect the privacy of data. Differential privacy is defined as follows:

**Definition 4.** *ε-Differential privacy [30]. For a given neighboring data set O and O', there is at most one record difference between them, and the number of different records is recorded as |OΔO'|, |OΔO'| indicates the number of records, where |OΔO'|= 1, given a random algorithm A and its defined and value domains Dom(A), Range(A), if algorithm A outputs any result D on data*

sets O and O' (D ∈ Range(A)) satisfies the following inequality, then A satisfies the ε-differential privacy, namely:

$$Pr[A(Q,O) \in D] \leq exp(\varepsilon) \times Pr[A(Q,Q') \in D] \tag{4}$$

The probability $Pr[\cdot]$ is controlled by the randomness of the algorithm and represents the risk of privacy disclosure, ε, which is the privacy budget, indicating the degree of the privacy budget. The smaller the value of ε, the higher the degree of privacy protection.

**Definition 5.** *Global sensitivity [30]. Define the proximity data sets O and O', for arbitrary query functions $f : O \rightarrow R^d$, then the global sensitivity of the query function is:*

$$\Delta f = \max_{O,O'} \|f(O) - f(O')\|_1 \tag{5}$$

*where, R represents the real number space mapped by the function, d represents the query dimension of function f, and $\|_1$ represents the sum of absolute values of each element.*

**Definition 6.** *Laplace noise [30]. Given data set O, its query function $f : O \rightarrow R^d$, if the output of algorithm A satisfies the following Equation (5), then algorithm A satisfies ε-differential privacy, i.e.,*

$$\widehat{f}(D) = f(D) + Laplace(\frac{\Delta f}{\varepsilon}) \tag{6}$$

*where, $Laplace(\frac{\Delta f}{\varepsilon})$ represents the added Laplacian noise. The Laplacian mechanism perturbs the real query results with noise satisfying the Laplace distribution so as to protect privacy.*

## 4. Multiuser Incomplete Preference Clustering Method Based on Association Rules

When initiating the *k* neighbor query for multiple users, the obtained user preferences are incomplete, and the multiuser preferences are different; thus, in order to resolve the problem, this section first optimizes the UFP-tree, proposes the one-way frequent pattern tree HUFP-tree, uses the tree to mine the frequent itemset to complete the incomplete preference of multiple users, further cluster the multi-query users according to the strong association rules mined by the frequent itemset, and finally group the query users according to the road network distance between the query users in the class cluster.

### 4.1. Multiuser Incomplete Preference Completion Method

This section proposes a completion algorithm for multiuser incomplete preference, which first mines a maximum-frequent itemset in user preference, so this section proposes a HUFP-tree frequent pattern tree based on the UFP-tree and then completes the multiuser incomplete preference by "0" and "1" according to the frequent itemset mining from this tree.

Before frequent itemset mining, this paper will perform "0" and "1" binarization on the multiuser preference of the same point of interest *k* neighbor query and mark the dimension as "1" when the query user's *k* neighbor query contains a certain attribute of the data object; otherwise, it is "0", and the attribute preference of multiple users is represented by a matrix. The completion criterion for incomplete preference is based on the length of the query user who prefers the most data object attributes. The definition of the frequent mode tree HUFP-tree is shown in Definition 7:

**Definition 7.** *HUFP-tree. HUFP-tree is a tree structure similar to the FP-tree, which consists of a frequent item header table and a HUFP-tree node with a null root node.*

**Definition 8.** *Frequent item header table. The frequent item header table consists of the item's sequence number $Item_{sn}$, the item's support count $Item_{cou}$, the $Item_{nowptr}$ pointer to the current node, and the $Item_{first}$ pointer to the child node where the sequence number appears for the first time.*

**Definition 9.** *HUFP-tree node. Each node consists of the item's support count Item$_{cou}$, the current node position Item$_{nowptr}$, a pointer to the parent node Item$_{parent}$, and a pointer to the sibling node with the same sequence number Item$_{bro}$, Item$_{now}$ consists of the item sequence number and hash function.*

**Definition 10.** *Frequent itemsets [1]. Let X be a sequence consisting of one or more items in the sequence{x1, x2, ..., xn}, if sup(X) supmin, the item set in X is said to be a frequent itemset, supminw is the minimum support for the itemset.*

**Definition 11.** *Maximal frequent itemsets [1]. Let X = {x$_1$, x$_2$, ..., x$_n$} is the itemset, S = {x$_i$, x$_j$, ..., x$_n$} is the frequent itemset, and S $\subseteq$ X, then $\forall$ x$_k$ $\in$ X and x$_k$ $\notin$ S, S $\cup$ x$_k$ is not a frequent itemset, so S is a maximum frequent itemset.*

The transaction set is shown in Table 1, assuming that the minimum support is 40%, and according to the above definition, the HUFP-tree is constructed as shown in Figures 3–5.

**Table 1.** Transaction data set 1.

| TID | Itemsets |
|---|---|
| 01 | {a, c, d, g, m, p} |
| 02 | {a, c, l, o} |
| 03 | {a, b, c, m, o} |
| 04 | {a, b, f, p, s} |

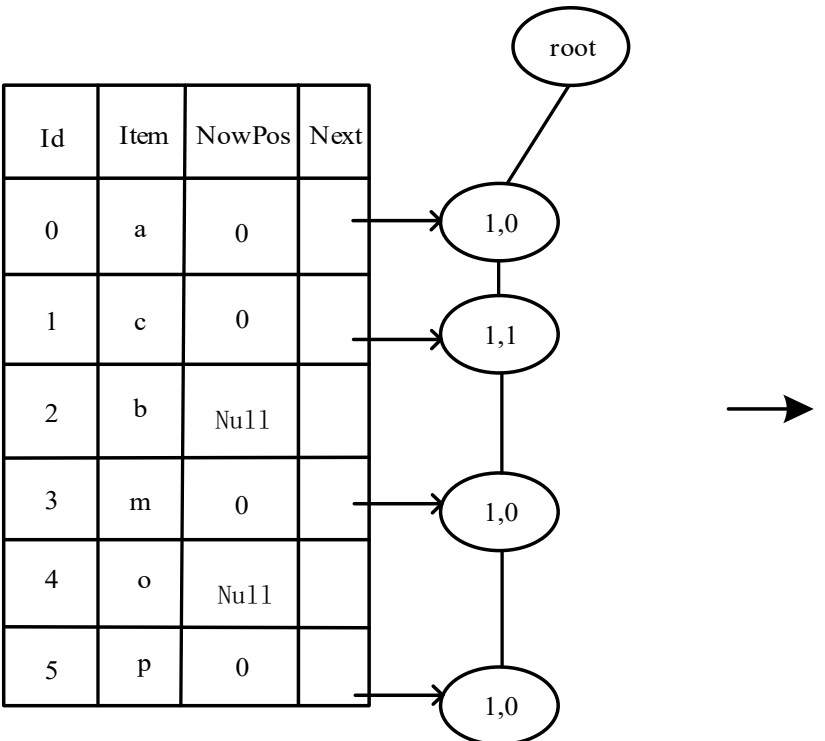

**Figure 3.** Add the first transaction(a, c, m, p).

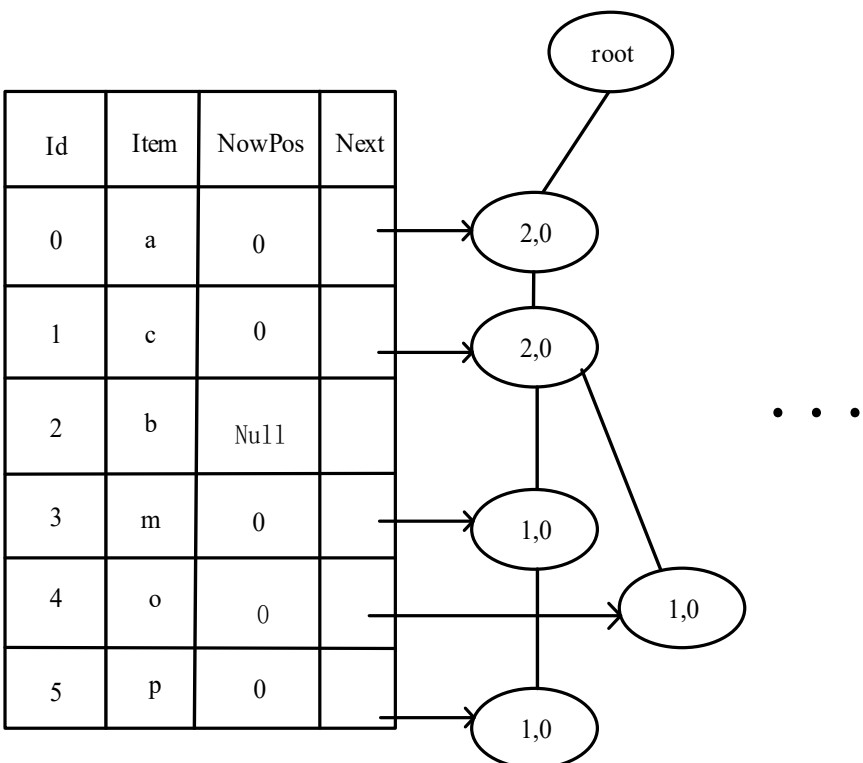

**Figure 4.** Add the second transaction(a, c, o).

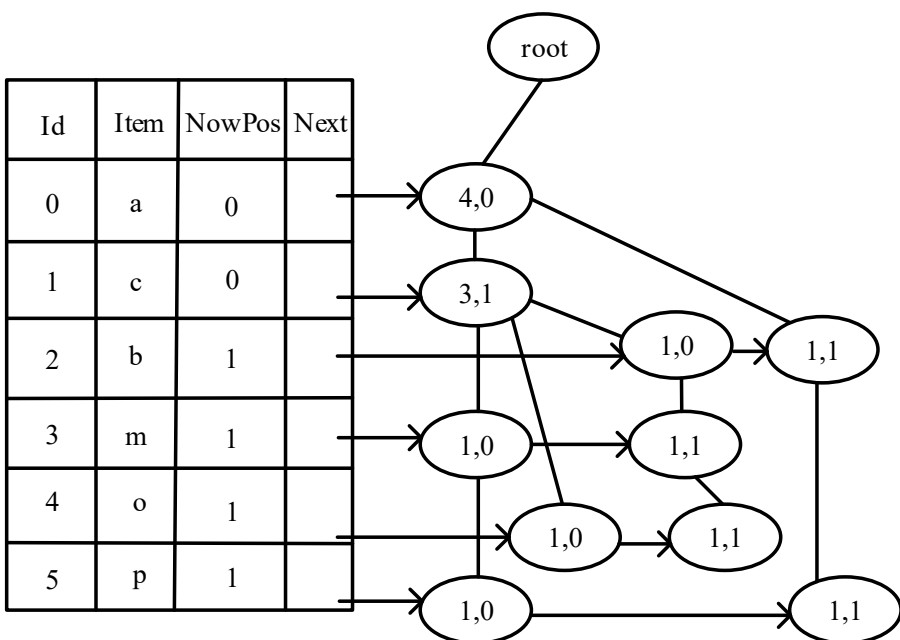

**Figure 5.** Construction of the HUFP tree.

According to Table 1, the construction process of Figure 5 is as follows: First, the transaction data set is scanned to obtain frequent 1 item set F(the number of occurrences of each item), as shown in Table 2. Because the minimum support is 40%, according to the existing support calculation formula, $4 \times 0.4 \approx 2$ can be obtained; that is, each item must appear at least twice. Therefore, d, f, g, s, and l, whose occurrence times are less than two, are deleted, and then the remaining items are sorted according to the decreasing support, as shown in Table 3. The transaction data set is then rearranged so that the items in each transaction are in decreasing frequency order, as shown in Table 4. After the preparation,

the root node needs to be created, and the first transaction (a, c, m, p) needs to be added, as shown in Figure 3. The second transaction (a, c, o) is shown in Figure 4. In addition, The tree adds the Item$_{nowptr}$ pointer to the current latest node corresponding to the ordinal number of the item header table; when the next scan to the node with the same ordinal number, the Item$_{bro}$ pointer in the node can directly point to the node pointed to by the Item$_{nowptr}$ pointer, saving the search process from the item header table to the latest node.

**Table 2.** The support of each item.

| a | b | c | d | f | g | l | m | o | p | s |
|---|---|---|---|---|---|---|---|---|---|---|
| 4 | 2 | 3 | 1 | 1 | 1 | 1 | 2 | 2 | 2 | 1 |

**Table 3.** Eligible items.

| a | b | c | m | o | p |
|---|---|---|---|---|---|
| 4 | 2 | 3 | 2 | 2 | 2 |

**Table 4.** Transaction data set 2.

| TID | Itemsets |
|---|---|
| 01 | {a, c, m, p} |
| 02 | {a, c, o} |
| 03 | {a, c, b, m, o} |
| 04 | {a, b, p} |

After the construction of the HUFP tree, in order to mine frequent itemsets, the following theorem is proposed in this section.

**Theorem 1.** *If $L = \{L_1, L_2, \ldots, L_n\}$ is the set of maximal frequent itemsets, $I = \{I_1, I_2, \ldots, I_n\}$ is a frequent itemset; then $\exists\, L_i \in L$ makes $I \subseteq L_i$.*

**Proof of Theorem 1.** *Let X be an itemset, and $\exists\, I$ is a frequent itemset; according to Definition 11, $\exists\, x_i \in X$ and $x_i \notin I$ makes $\{x_i\} \cup I$ a frequent itemset.* □

**Theorem 2 ([1]).** *A superset of infrequent itemsets must not be a frequent itemset.*

**Corollary 1.** *A maximum-frequent itemset is a frequent itemset that qualifies for no superset in each frequent itemset.*

**Corollary 2.** *Let $L = \{l_1, l_2, \ldots, ln\}$ is a maximal frequent itemset, then the itemset composed of $\forall\, l_i \in L$ must be a frequent itemset.*

**Theorem 3.** *Let $X = \{x_1, x_2, \ldots, x_n\}$ is a set of items, $\forall\, x_i \in X$, if $sup(x_i) < sup_{min}$, then the itemset $x_i$ is pruned.*

**Proof of Theorem 3.** *According to Definition 10, $sup_{min}$ is the minimum support degree of itemset, $sup(X) \geq sup_{min}$ is a frequent itemset; therefore, it can be pruned directly when $sup(x_i) < sup_{min}$.* □

**Theorem 4.** *On the HUFP-tree, the node with sequence number i is the leaf node, and the support count on its path to the root node is equal to the support count of sequence i on the path, and if the sequence i has a brother node, the frequency count on its common path is the sum of the support of the sequence i node.*

**Proof of Theorem 4.** *Let $X = \{x_1, x_2, \ldots, x_i\}$ is the set of nodes passing through the path from node i to the root node, count for the support count of each node, according to the definition and construction of the HUFP-tree, $\forall\, x_j \in \{X - \{x_i\}\}$, count($\{x_i\}$) $\leq$ count($\{X - \{x_i\}\}$), that is, the leaf node $x_i$ is the node with the smallest support degree count on the path, so the support degree count on the path is the support count of node $x_i$, and because the node with the ordinal number $x_i$ represents the same itemset, therefore, the common node in all paths with the ordinal number $x_i$ as the leaf node is the sum of the support of the ordinal i node. Certified.* □

According to Tables 3 and 4, we first consider p to obtain the conditional pattern basis (that is, all path branches with p as a node) and set the number of nodes it passes through to 1 (because nodes p, b, and a only contribute to e once, and others similarly), {a, c, m :1} and {a, b :1} are obtained. Using the conditional pattern base, the conditional HUFP-tree is constructed here.

Figure 6 shows the frequent itemset construction process 1 for p; Figure 7 shows the frequent itemset construction process 2 for p.

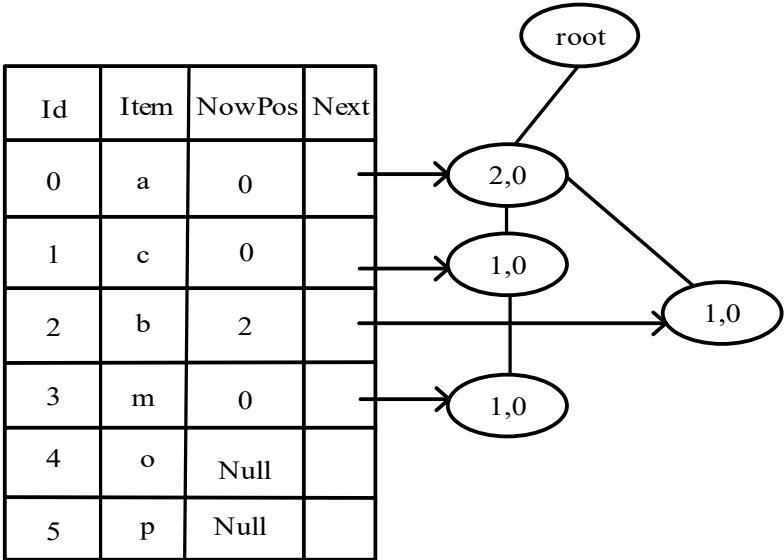

| Id | Item | NowPos | Next |
|----|------|--------|------|
| 0 | a | 0 | |
| 1 | c | 0 | |
| 2 | b | 2 | |
| 3 | m | 0 | |
| 4 | o | Null | |
| 5 | p | Null | |

**Figure 6.** Frequent itemset construction process 1 for p.

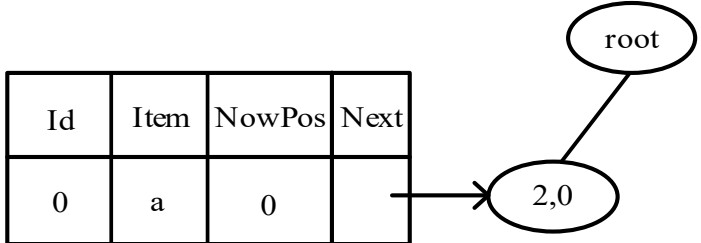

| Id | Item | NowPos | Next |
|----|------|--------|------|
| 0 | a | 0 | |

**Figure 7.** Frequent itemset construction process 2 for p.

This paper has declared that the minimum support is 2, and the support of c, b, and m in the tree obtained by p is less than 2, so the combination with e cannot reach the support of 2, so it is removed:

Then the remaining a is satisfied, and the frequent item set combined with p is: {a, p:2}.

According to the above definition and theorem, frequent itemsets that meet the threshold of support count are frequent itemsets, and incomplete preference of multiple users can be completed after the frequent itemsets of user preference are mined.

For example, the incomplete preference of query user $q_1$ is (1, -, -, -), "-"indicates a lack of preference, assuming that the frequent itemset with a preference of 1 in the first dimension mined is (1, 1, 0, 0), then the preference after $q_1$ completion is (1, 1, 0, 0).

Further, we give the multiuser incomplete preference completion algorithm PCAR, as shown in Algorithm 1. Firstly, the 1 itemset that does not meet the threshold is deleted to obtain the user preference of the frequent 1 itemset, then the HUFP-tree is constructed, and the frequent itemset is excavated to obtain the extremely frequent itemset. Finally, the multiuser incomplete preference is completed based on the extremely frequent itemset.

---

**Algorithm 1**: Multiuser incomplete preference completion algorithm (PCAR)

---

**Input**: Transaction set $D = \{D_1, D_2, \ldots, D_n\}$, $d_i$ is the item set in $D_n$, and the minimum support threshold $sup_{min}$.
**Output**: Preference after multiuser completion.
begin
1:   $candi \leftarrow \varnothing$, $candiRes \leftarrow \varnothing$;
2:   for each $d_i \in D_i$, calculate $sup(D_i)$;
3:     if $sup(d_i) > sup_{min}$;
4:       $candi \leftarrow d_i$;
5:     end if;
6:   else
6:       delete $d_i$
7:     end else;
8:   end for;
9: Sort the items in *Candi* in descending order of frequency count and get the corresponding ordinal number;
10: creating a root Noderoot = null;
11: create a frequent item header table;
12: for $d_i$ in *candi*;
13:     if the item header table ordinal number is equal to the node $d_i$ ordinal number; &&Item$_{first}$==null;
14:       count(Item$_{cou}$) + 1, Item$_{now}$ saves the $d_i$ location, Item$_{nowptr}$ and Item$_{first}$ point to $d_i$, Item$_{parent}$ point to $d_i$'s
parent;
15:     end if
16:     if the ordinal number of the header table is equal to the ordinal number of the node $d_i$ &&Item$_{firs}$!==null
17:       count(Item$_{cou}$) + 1, Item$_{now}$ saves the $d_i$ location, Item$_{parent}$ point to $d_i$'s parent, Item$_{nowptr}$ and Item$_{bro}$ point to $d_i$;
18:     end if;
19: end for
20: for $d_i$ in *candi*;
21:     sets the count of nodes in the $d_i$ to root path as an item, the count value of the serial number corresponding to $d_i$ in the header table;
22:     if count value < $sup_{min}$;
23:       delete the corresponding node to get the frequent itemset;
24:   end if
25: end for
26:   $candi\_f \leftarrow$ Merging all frequent itemsets results in extremely frequent itemsets;
27: for $D_i$ in D;
28:   if $D_i$ has missing dimensions
29:     for X in *candiRes*
30:         completion $D_i$;
31:       end for
32:   end if
33: end for
end

---

The algorithm first prunes the 1 itemset in transaction set D that is less than the support threshold to obtain a frequent 1 itemset (rows 1–9). Then build the HUFP tree on it (lines 10–20). Then, based on the existing UFP tree mining method and the HUFP-tree and related theorems mentioned in this section, this section mines the data objects by pruning and frequent itemsets and then merges the frequent n itemsets of each data to obtain the extremely frequent itemset of the item (lines 21–27). Finally, the missing dimensions of multiuser preference can be completed directly from the extremely frequent itemset (lines 28–34).

A description of user preferences is shown in Tables 5 and 6.

**Table 5.** Attributes of the cafe.

| Cafe | Taste | Environment | Service | Per Capita Price | Road Network Distance/m |
|------|-------|-------------|---------|------------------|-------------------------|
| $p_1$ | 1 | 2 | 2 | 70 | 20 |
| $p_2$ | 1 | 1 | 2 | 70 | 50 |
| $p_3$ | 3 | 3 | 1 | 80 | 5 |
| $p_4$ | 4 | 2 | 2 | 50 | 10 |
| $p_5$ | 1 | 1 | 4 | 65 | 30 |

**Table 6.** User preference.

| Cafe | Taste | Environment | Service | Per Capita Price | Road Network Distance/m |
|------|-------|-------------|---------|------------------|-------------------------|
| A | ☆☆ | - | ☆ | - | ☆ |
| B | ☆☆ | ☆ | ☆ | ☆ | ☆ |
| C | ☆ | ☆☆ | ☆ | - | - |
| D | ☆ | - | - | - | - |
| E | ☆ | - | ☆ | ☆ | ☆☆ |

**Example 1.** *A group of students needs to select some cafes. The data information on cafes is shown in Table 5. Among them, the three attributes of taste, environment, and service are displayed in the form of numerical scores. The larger the value, the better, and the missing data are represented by "-". Group members A, B, C, D, and E put forward different query requirements according to their own preferences, respectively, and gave the attributes they thought were the most important, as shown in Table 6.*

User A: To inquire about a cafe with good taste, good service, and a close road network, taste is preferred.

User B: To query cafes with good taste, good environment, good service, low per capita price, and close road network distance, the taste is preferred.

User C: Inquire about good taste, good environment, good service cafe, and environment priority.

User D: To query the cafe with good taste, the taste is preferred.

User E: Inquire about the cafe with good taste, good environment, good service, low price per capita, and close road network distance; road network distance is preferred.

Assuming that the minimum support is 40%, then taste and service must be frequent itemsets. Without considering other attributes, the data after completion for D is (1, 1, 1, 0, 0), where "1" means that there is a preference for this attribute, and "0" means that there is no preference for this attribute. After obtaining the completed data of all users, multiusers can be clustered by a user clustering algorithm.

*4.2. Multiuser Clustering Algorithm Based on Association Rules*

After mining the frequent itemset for each preference dimension in the multiuser preference in Section 4.1, this subsection obtains the positive correlation coefficient of the association rule according to the frequent itemset, proposes a distance-preference similarity measurement method based on the positive correlation coefficient and the road network distance between multiple users, and then clusters users with large similarity according to this method, so as to ensure that users in the same class cluster have a closer road network distance and extremely similar preference. The similarity measure is shown in Equation (7).

$$compare(q_i, q_j) = \sqrt{\sum_{i=1}^{n} \frac{pref_{w_i w_j}}{d(q_i, q_j)^2}} \tag{7}$$

$$pref_{w_i w_j} = \frac{sup(w_i \cup w_j) - sup(w_i)sup(w_j)}{\sqrt{(1 - sup(w_i))(1 - sup(w_i))sup(w_i)sup(w_j)}} \tag{8}$$

Among them, $pref_{w_i w_j}$ is the positive correlation coefficient of each user's main preference, $w_i$ represents the support degree of the key preference of the query user $q_i$, and $w_j$ represents the support degree of the key preference of the query user $q_j$. The main preference is determined by the largest preference of the original support count value in the corresponding item header table in each query; user preference is the user's key preference.

According to the definition of association rules, when association rule $A \rightarrow B$ is strongly correlated, the occurrence of item set $A$ will lead to a greater probability of item set $B$. Therefore, according to the above description and similarity measurement method, the multiuser clustering algorithm in this section is correct, as shown in Theorem 5:

**Theorem 5.** *If $Q = \{q_1, q_2, \ldots, q_n\}$ represents the query user and $C = \{c_1, c_2, \ldots, c_n\}$ represents the cluster, then based on the similarity measure method, for $\forall q_i \in Q$, there is $q_i \in c_i$.*

**Proof of Theorem 5.** *If $con(A \rightarrow B) = 1$, then the number of occurrences of $A \cup B$ is equal to the number of occurrences of $A$, so the occurrence of $A$ will lead to the occurrence of $B$. In the similarity measurement method, assuming that the road network distance between users is unchanged, then when the positive correlation coefficient is larger, the higher the similarity between users, so it is easier to be divided into the same cluster, ensuring that users in the same cluster have high cohesion. Certified.* □

The main idea of the multiuser clustering algorithm based on association rules proposed in this section is as follows: this section first obtains the association rules according to the frequent term set and the *pref* coefficient and proposes a similarity measurement method according to the association rules and the road network distance between users, and then the method performs clustering operations on multiple users. Further, this section presents a multiuser clustering algorithm based on association rules, as shown in Algorithm 2.

The multiuser clustering algorithm based on association rules first traverses frequent itemsets to obtain positive correlation association rules (lines 1–13). Then the algorithm obtains the correlation coefficient between query users according to the key preference of the query user (lines 14–16) and further calculates the similarity measure between query users according to the correlation coefficient and the road network distance between users. If the value is greater than the given threshold, then multiple query users are divided into the same class cluster; otherwise, it is self-contained (lines 17–24).

---

**Algorithm 2:** Multiuser clustering algorithm based on association rules (MCR)

---

**Input**: Query user $Q = \{q_1, q_2, \ldots, q_i\}$, frequent itemsets *candiRes*
**Output**: multiuser clustering $C$.
begin
1:   $ASSR \leftarrow \varnothing$, $C \leftarrow \varnothing$, $i = 0$;
2: for $X$ in *candiRes*
3:    if $con(A \rightarrow B) \geq 1$ then
4:       $ASSR \leftarrow Pr \cup (A \rightarrow B)$
5:    end if
6: end for
7: for $F$ in $ASSR$
8:    find the subset $\hat{B}$ of itemset $B$
9:    if $con(A \rightarrow \hat{B}) \geq 1$ then
10:    $ASSR \leftarrow ASSR \cup (A \rightarrow \hat{B})$
11:   end if
12: end for
13:  $ASSR \leftarrow ASSR - (A \rightarrow \hat{B})$;
14: for $j$ in $Q$
15:    Get the key preference of each query user based on the preference of each query user and the corresponding item header table $A_i$, $B_j$;
16:    Calculating positive correlation coefficient according to association rules $pref_{A_i B_j}$;
17:    Calculate the similarity measure between every two query users $compare(A_i, B_j)$;
18: end for
19: if $compare(A_i, B_j) \geq compare_{min}$
20:    $C_i \leftarrow A_i \cup B_j$
21: end if
22: else $C_j \leftarrow B_j$
23: $i \leftarrow i + 1$
24: return $C$;
end

---

## 5. Multiuser Preference *K*-Nearest Neighbor Query Algorithm Based on Differential Privacy in Road Network

Section 3 clustered query users according to the similarity measurement method after completing the incomplete preference of multiple users so that the query users in the class cluster obtained on the basis of considering preference and distance have great similarity, which provides convenience for subsequent $k$ neighbor queries. Therefore, this section mainly studies the $k$-nearest neighbor query process, first reducing the data object in the distance dimension of the road network, then refining the data object that does not meet the preference of multiple users, then performing the $k$-nearest neighbor query for the classified query users, and finally using differential privacy technology to protect the data results from being published.

### 5.1. Reduced Data Object

This section reduces the data object according to the road network distance between the query object and the data object, first defines the minimum circumscribed circle of the Voronoi polygon to determine the query center and query area of each class cluster, and then reduces the data object according to the theorem [33] proposed by the nature of the road network Voronoi graph.

**Definition 12.** *Minimum circumcircle of the cluster. If $c_i$ is the ith cluster, $c_i = \{p_1, p_2, \ldots, p_i\}$, the smallest circle of all data objects $p_j$ $(1 \leq j \leq i)$ that contain $c_i$ is the smallest circumcircle of the cluster.*

When the number of query objects in a class cluster is 1, it is itself the query center; when the number is 2, the query center is the center of the distance between the two data

points; when the number is 3, the query center is the center of the smallest circumscribed circle of the class cluster, and the query area is a circle with the midpoint as the center and the distance from the midpoint to the query object is the radius. As shown in Figure 8, there is only one query object in cluster $s_1$, $c_1$ is the query area to point $p$; cluster $s_2$ has two data objects, then $o_2$ is the query center, $c_5$ is the query area, there are three data objects in cluster $s_3$, the query center is $o_1$, $c_3$ formed by $o_1$ and $p$ is its query area. Further, this paper uses the road network Voronoi diagram and the query area of the query object to reduce the data object.

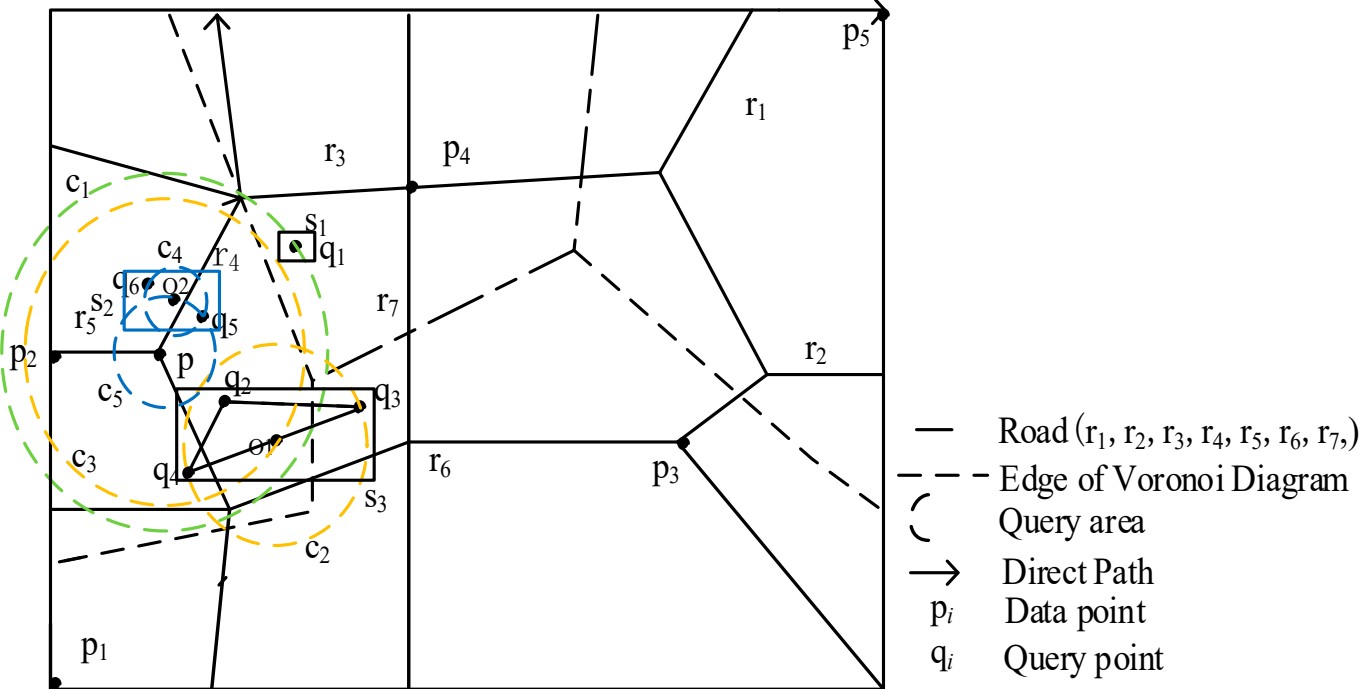

**Figure 8.** Query area example.

**Theorem 6.** *In the Voronoi graph of the road network constructed by the data object, if the query object p belongs to the query area CH($c_i$), the data points outside the circle domain tangent to CH($c_i$) of the data object are dominated by p, so the data points outside the region can be reduced.*

**Theorem 7.** $\exists\, p_i, p_j \in P$, *if the Voronoi unit VC(p) of the road network where $p_i$ is located intersects with the query area CH($c_i$), if $p_j$ is a data point on any road network that does not intersect with $p_i$, then $p_j$ is dominated by $p_i$, so $p_j$ is pruned.*

**Theorem 8.** $\exists\, p \in P$, *if the road network where p is located intersects the boundary of the given query convex hull region, p is a candidate data set.*

### 5.2. Refining Data Objects

In order to refine the reduced data objects, this section first proposes the concept of clustering preference order chain based on the key preference in each cluster and then proposes δ-SNS domination, which compares the dimensions in the data object from back to front according to the order chain order. Then it compares the traditional domination relationship between the dimensions that are not relaxed so as to achieve the purpose of pruning the data objects with weak dominance in each dimension.

**Definition 13.** *Clustering preference order chain. Let H = {$h_1$, $h_2$, ..., $h_n$} be the preference set in cluster $C_i$. According to the support degree in the frequent item header table, the ordered set*

*sorted from large to small is called the clustering preference order chain, which is expressed as $h_1 < h_2 < \ldots < h_n$, where the importance is decreasing, and $h_1$ is the main preference.*

**Definition 14.** *$\delta$-SNS dominance. Let $O = \{o_1, o_2, \ldots, o_n\}$ be the set of d-dimensional data objects. Assuming that $o_1$ dominates $o_2$, we need to satisfy $o_2.[h_i] - o_1.[h_i] \leq \delta_i$. Among them, according to the cluster preference order chain, the dominance comparison is carried out from back to front, and i is the different relaxation granularity. When the preference importance is higher, i is smaller, indicating that the degree of relaxation allowed on the preference is smaller.*

Based on the clustering preference order chain and $\delta$-SNS dominance, the process of dominance comparison between data objects is as follows: according to the order of the tail of the clustering preference order chain to the head, the data object is compared with $\delta$-SNS dominance. It is assumed that one dimension t is relaxed each time, the dimension before t adopts the traditional strict dominance strategy, and the dimension t and the later relaxed dimensions still adopt $\delta$-SNS dominance comparison. In order to reduce the number of comparisons between the dominance relations between data objects, we give Theorem 9.

**Theorem 9.** *Let P be a data set with n dimensions, $p_1$, $p_2 \in P$, then $p_1$ also dominates the data object $p_2$ in the first n-m dimension, so $p_2$ can be pruned.*

**Proof of Theorem 9.** *If $p_1$ dominates the data object $p_2$ in the reciprocal $\delta$-SNS, then the reciprocal m-dimension adopts a relaxed granularity for domination comparison, so the first n-m dimension adopts the strict mode, and in the non-strict mode p2 can dominate p2, then $p_2$ is more dominated by $p_1$ in the strict mode, so it can be pruned.* □

This section further gives the reduction and refinement algorithm of data objects in the road network, as shown in Algorithm 3.

The algorithm first establishes a Voronoi graph of the road network for the set of data objects, then traverses the class clusters, determines the query center, query area, and the minimum circumscribed circle composed of the data object and query center of each class cluster (lines 1–6). It then subtracts the data objects according to Theorems 6, 7, and 8 (lines 7–19) and finally uses the $\delta$-SNS domination comparison method and Theorem 9 mentioned in this section to refine the reduced data set to obtain the candidate result set (lines 20–28).

After obtaining the candidate result set of each class cluster through Algorithm 3, in the final query stage, the road network distance $d(\breve{c}_i, p)$ between the query center point $\breve{c}_i$ of each cluster and each data object $p$ in the candidate result set is calculated respectively, and it is arranged in the order from small to large, and finally, the first $k$ data objects satisfying the conditions are obtained to form the result set.

After querying the result set of each cluster, this topic adds Laplace noise to the query result set to protect the privacy of the data result when the data result is published. The total differential privacy budget is as follows: this article allocates the privacy budget parameters based on the weight ratio of each cluster, where the weight of each cluster is the support degree of the main preference of each cluster. Assuming that there are $n$ classes, the privacy budget assigned to each class is $\varepsilon_j = \dfrac{\varepsilon w_j}{n \sum\limits_{i=1}^{n} w_i}$. The Laplacian noise added for each class is:

$$lap(\frac{\Delta f}{\varepsilon_i}), \ (1 \leq i \leq n) \tag{9}$$

Here, $\Delta f$ is the global sensitivity, where the global sensitivity is $\Delta f = k$, ($k$ is the number of result sets).

---

**Algorithm 3:** Reduction and refinement algorithm of data objects in road network (RAR)

---

**Input**: Data object set $P$, road segment set $R$, cluster set $C = \{c_1, c_2, \ldots, c_m\}$, $ci = \{p_1, p_2, k, p_i\}$;
**Output**: Candidate result set *CondiResSet*;
begin
1:   Construct a Voronoi diagram of the road network from road segments and data objects as generated points;
2:   *ResSet* $\leftarrow \varnothing$;
3:   for $c_i$ in $C$ then
4:   Determining the minimum circumcircle of $c_i$ to form the query region $CH(c_i)$;
5:   if $p$ in $CH(c_i)$ then
6:      $AR \leftarrow$ circle($p, \breve{c}_i$);/*gets the minimum circumscribed circle of the query center and data objects*/
7:   end if
8:   if $p$ in $AR$ then
9:      *ConSet* += $p$;/*theorem 6*/
10:   end if
11:   if $CH(c_i)$ and $V_{\mathrm{NVP}}(p)$intersect then
12:   *ConSet* $\leftarrow$ *ConSet* + $p$;/*theorem 7*/
13:   end if
14:   else
15:   $r \leftarrow CH(c_i)$ sections intersecting $R$
16:   if $p$ in $r$ then
17:      *ConSet* $\leftarrow$ *ConSet* + $p$;/*theorem 8*/
18:   end if
19:   end else
20:   for $p_i$ && $p_j$ in *ConSet* then
21:   $\delta$-SNS dominance comparison of $p_i$ and $p_j$;
22:   if $p_i$ dominates $p_j$ then
23:      *PreSet* $\leftarrow p_j$;/*theorem 9*/
24:   end if
25:   end for
26:   *CondiResSet* $\leftarrow$ *ConSet-PreSet*;
27:   return *CondiResSet*;
end for
end

---

Based on the above discussion, this section further gives the multiuser preference $k$-nearest neighbor query algorithm based on differential privacy in the road network, as shown in Algorithm 4.

---

**Algorithm 4:** Multiuser preference $k$-nearest neighbor query algorithm based on differential privacy in road network (PIPKQ)

---

**Input**: Candidate result set *CondiResSet*, cluster set $C = \{c_1, c_2, \ldots, c_m\}$, privacy budget $\varepsilon$;
**Output**: Noisy result set *ResSet*;
begin
1:   for each *CondiResSet* then/*traverse the candidate result set of each cluster*/
2:   for $p$ in *CondiResSet* then
3:      candidate $d(\breve{c}_i, p)$
4:      sort(d($\breve{c}_i, p$));
5:   *ResSet* $\leftarrow$ the first $k$ shortest distance data objects;
6:   end for
7:   Assign privacy budget parameters to each cluster $\varepsilon_j = \varepsilon w_j / n \sum\limits_{i=1}^{n} w_i$;
8:   Calculate the noise added in each cluster $lap(\frac{\Delta f}{\varepsilon_i})$;
9:   end for;
10: return noisy result set *ResSet*;
end

In the multiuser preference *k* neighbor query algorithm based on differential privacy in the road network, the distance from the data object *p* in each candidate result set to the query center of its corresponding class cluster is first calculated and arranged in order from smallest to largest. The first *k* data objects are selected as the result set of the corresponding class cluster (lines 1–6), and finally, noise is added to each result set (lines 7–9).

*5.3. Privacy Analysis*

This section mainly analyzes whether the algorithm satisfies $\varepsilon$-differential privacy, and in order to verify that the PIPKQ algorithm satisfies $\varepsilon$-differential privacy, Theorem 10 is given in this section.

**Theorem 10.** *Algorithm PIPKQ satisfies $\varepsilon$-differential privacy.*

**Proof of Theorem 10.** *Because the algorithm allocates the privacy budget according to the number of clusters and the proportion of weights, let the neighbor data sets be D and Dı, ALG represents the PIPKQ algorithm, $A_{ALG}$ represents the set of all possible outputs of the algorithm ALG, $S_M$ is a subset of any $A_{ALG}$, ce(x) represents the query results after using the PIPKQ algorithm, ce(D) and ce(Dı) represent the real query results. According to the differential privacy distribution function, there is*

$$
\begin{aligned}
A_{ce}[ALG(D) \in S_{ALG}] \quad &= A_{ce}[lap[1/\varepsilon_1] + lap[1/\varepsilon_2] + \cdots + lap[1/\varepsilon_n] \\
&= ce(x) - ce(D)] \\
&= (\varepsilon_1/2)\exp(-\varepsilon_1|ce(x) - ce(D)|) \\
&\quad (\varepsilon_2/2)\exp(-\varepsilon_2|ce(x) - ce(D)|) \cdots (\varepsilon_n/2)\exp(-\varepsilon_n|ce(x) - ce(D)|)
\end{aligned}
$$

$$
\begin{aligned}
A_{ce}[ALG(Dı) \in S_{ALG}] \quad &= A_{ce}[lap[1/\varepsilon_1] + lap[1/\varepsilon_2] + \cdots + lap[1/\varepsilon_n] \\
&= ce(x) - ce(Dı)] \\
&= (\varepsilon_1/2)\exp(-\varepsilon_1|ce(x) - ce(Dı)|) \\
&\quad (\varepsilon_2/2)\exp(-\varepsilon_2|ce(x) - ce(Dı)|) \cdots (\varepsilon_n/2)\exp(-\varepsilon_n|ce(x) - ce(Dı)|)
\end{aligned}
$$

□

According to the definition of sensitivity: $\|ce(x) - ce(Dı)\|_1 \leq \Delta f = k$, (*k* is the number of results), so

$$
\begin{aligned}
&A_{ce}[ALG(D) \in S_{ALG}] / A_{ce}[ALG(Dı) \in S_{ALG}] \\
&= \exp(-\varepsilon_1|ce(x) - ce(D)| - \varepsilon_2|ce(x) - ce(D)| - \cdots - \varepsilon_n|ce(x) - ce(D))|) \\
&/ \exp(-\varepsilon_1|ce(x) - ce(Dı)| - \varepsilon_2|ce(x) - ce(Dı)| - \cdots - \varepsilon_n|ce(x) - ce(Dı)|) \\
&= \exp(\varepsilon_1(|ce(x) - ce(Dı)| - |ce(x) - ce(Dı)|) + \varepsilon_2(|ce(x) - ce(Dı)| \\
&\quad -|ce(x) - ce(Dı)|) + \cdots + \varepsilon_n(|ce(x) - ce(Dı)| - |ce(x) - ce(Dı)|)) \\
&\leq \exp(\varepsilon_1\|ce(x) - ce(Dı)\|_1 + \varepsilon_2\|ce(x) - ce(Dı)\|_1 + \cdots + \\
&\quad \varepsilon_n\|ce(x) - ce(Dı)\|_1) \\
&\leq \exp[(\varepsilon_1 + \varepsilon_2 + \cdots + \varepsilon_n)n] \\
&= \exp(\varepsilon n)
\end{aligned}
$$

*5.4. Usability Analysis*

This section theoretically analyzes the availability of the PIPKQ algorithm using $(\alpha,\delta)$-useful technology.

**Definition 15.** *(α,δ)-useful. Given the data set O in the road network, suppose that the k nearest neighbor query algorithm in the road network is represented as B, and the result of algorithm B after adding noise to the data set O is expressed as O′, that is B(O) = O′, if the following formula is satisfied, algorithm B satisfies (α,δ)-useful.* $A_{ce}[|B(O\prime) - B(O)| \le \lambda] > 1 - \delta$.

**Theorem 11.** *PIPKQ algorithm satisfies the requirements* $\left( \frac{\ln(l/\delta)l}{(\varepsilon_1 \cdot \varepsilon_2 \cdot L \cdot \varepsilon_l)}, \delta \right)$, *where a is the privacy budget for each cluster.*

**Proof of Theorem 11.** *Let COL = {$o_1$, $o_2$, L, $o_k$} be the set of query results, and RES = {$o_1$′, $o_2$′, L, $o_k$′} be the set of noised results, the sum of the Laplacian noises added to each category is denoted by* $n_i (1 \le I \le k)$, *then* $O_i' = O + n_i$, *it is known by definition:* $|RES - COL| = \left| \sum_{i=1}^{k} O_i' - \sum_{i=1}^{k} O_i \right| = \left| \sum_{i=1}^{k} n_i \right| \le \sum_{i=1}^{k} |n_i| \le \lambda$, *in order to make the algorithm PIPKQ achieve the purpose of usability, let* $d = 1/(\varepsilon_1 \cdot \varepsilon_2 \cdots \varepsilon_k)$, *and is available from the literature* [34], $A_{ce}(|RES - COL| \le \lambda) = A_{ce}\left( \sum_{i=1}^{k} |n_i| \le \lambda \right) \ge 1 - \exp\left( \frac{-\lambda}{kd} \right)^k \ge 1 - k \times \exp\left( \frac{-\lambda}{kd} \right)$, *if* $k \times \exp(-\lambda/kd) \ge \delta$ *then* $A_{ce}(|RES - COL| \le \lambda) \le 1 - \delta$, *so* $(k \times \ln(k/\delta)/(\varepsilon_1 \cdot \varepsilon_2 \cdots \varepsilon_k)) \le \lambda$. □

## 6. Experiment Analysis

In the field of *k* nearest neighbor query, aiming at the problem of multiuser incomplete preference, this paper proposes a multiuser incomplete preference *k*-nearest neighbor query algorithm based on differential privacy in road network (PIPKQ). The algorithm first excavates the frequent itemset based on the HUFP tree, completes the incomplete preference according to the frequent itemset and obtains the positive correlation coefficient, proposes a similarity measurement method based on the positive correlation coefficient and road network distance, and clusters the query objects accordingly, the next step is to reduce and refine the data objects, then perform *k* neighbor query on the refined data, and finally add Laplace noise to the query results.

The experimental environment is: Microsoft Windows 10 (64-bit), Core(TM)i3-4005U CPU@1.70GHz processor, running memory is 8 GB, and the programming language is Java.

The experimental data comes from the open street map of an area in Atlantic City, the United States. The data set includes 41,905 data objects and 92,548 edges. In order to facilitate the experiment, the experiment makes appropriate adjustments to the query object and a data object, randomly selects 100 nodes to form a query set, the number of dimensions of the data object is 6~10, and each dimension is represented by a numerical value, and randomly generated between [0, 20].

This paper designs a comparative experiment in two aspects because most of the privacy issues in multiuser preference queries in the existing literature protect the location of data objects, so the comparison algorithm in this paper is mainly divided into two aspects. First, in terms of the *k*-nearest neighbor query to meet the multiple users' incomplete preference in the road network environment, since the paper studies the incompleteness of multiuser preferences and the preference of users in the *k*-nearest neighbor query in the existing road network environment is deterministic and complete data. Therefore, the QI-SCSA algorithm [31] and the algorithm anytimeQE-approx algorithm [35] are slightly improved (the missing data is completed) and compared with the PCAR method in this paper in terms of data missing rate. After completing the data by PCAR algorithm, the PIPKQ algorithm in this paper can be compared with the BA algorithm [12], PQA algorithm [13] and DSAS algorithm [36] (extended DSAS algorithm, scored data points meeting different user requirements, and returned *k* Skyline result sets from high to low according to data point scores, and the extended algorithm was called EDS) in terms of *k* value, number of POIs, and the number of query objects. To ensure the validity of the experiment, we took an average of 30 queries for analysis. Secondly, in the aspect of protecting the privacy of query results, the relative errors of the differential privacy budget allocation algorithm (DPBA),

arithmetic sequence allocation algorithm, and arithmetic sequence allocation algorithm [30] are compared and analyzed.

Among them, to compare the experiment's reliability with the PQA algorithm and the BA algorithm, the PIPKQ algorithm does not consider the privacy protection of the query results when making comparisons. In terms of privacy protection, it does not take into account the incomplete preference of multiple users in the road network for *k* neighbor queries.

### 6.1. Algorithm Comparative Analysis

In this section, we verify the feasibility and efficiency of the proposed algorithm through four experiments. In this paper, we evaluated our method in terms of the missing data rate, *k*-value, number of data points, and number of query objects, respectively. The parameters of the *k* value, number of data points, and number of query objects are shown in Table 7.

**Table 7.** Experiment parameter settings.

| Parameters | Default | Range |
|---|---|---|
| *k* | 10 | 5, 10, 15, 20, 25, 30, 35 |
| The data point number | 500 k | 50 k, 100 k, 500 k, 1 M, 1.5 M, 2 M |
| The query object number | 10 | 5, 10, 15, 20, 25, 30, 35 |

**Experiment 1.** *The impact of the data missing rate on execution time. In this experiment, the number of query objects is 100, and the dimension of data objects is 8. In the experiment, PCAR, QI-SCSA, and anytime QE-approximatex algorithms are compared. As the data loss rate increases, the changes in the three algorithms are shown in Figure 9.*

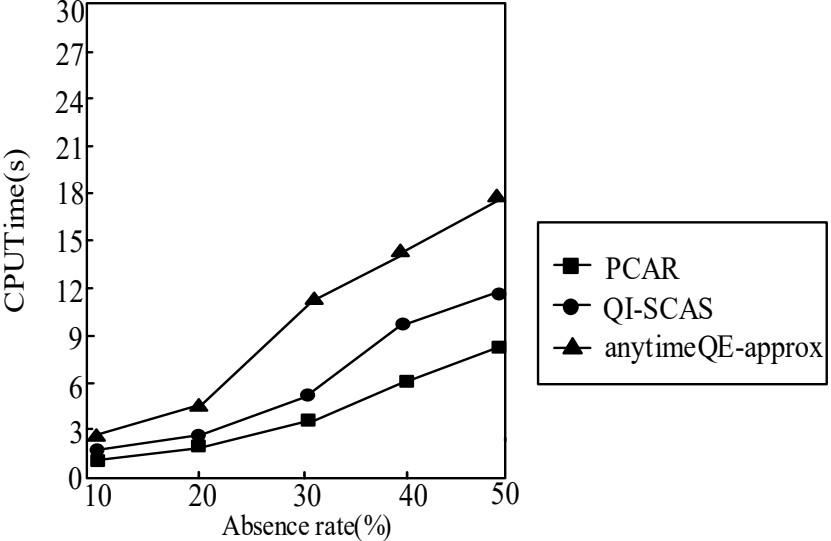

**Figure 9.** The impact of the data missing rate on execution time.

As can be seen from Figure 9, the execution time of the three algorithms increases with the increase of the data missing rate, but the PARGUE algorithm increases more slowly than the execution time of the QI-SCSA and anytimeQE-approx algorithms. This is because the QI-SCSA algorithm uses an iterative and heuristic strategy. The anytimeQE-approx algorithm iterates through each class in descending order according to the obtained probability and obtains the probability that a preference is queried, so it takes a long time. The PARGUE algorithm performs frequent itemset mining according to the existing preference of the query object to obtain the association relationship between the data and then completes the missing data. As the deletion rate gradually increases, the time

required for data completion increases, but the number of items in the frequent itemset also decreases, so the time increase is relatively slow.

**Experiment 2.** *The impact of the k-value on the performance of the algorithm. When the number of query objects is 100, the number of data objects is 10,000, the dimension of interest of each query object is 8, and the k value increases from 1 to 30, this paper compares and analyzes the PIPKQ algorithm, PQA algorithm, and BA algorithm in terms of CPU execution time and I/O cost, as shown in Figure 10:*

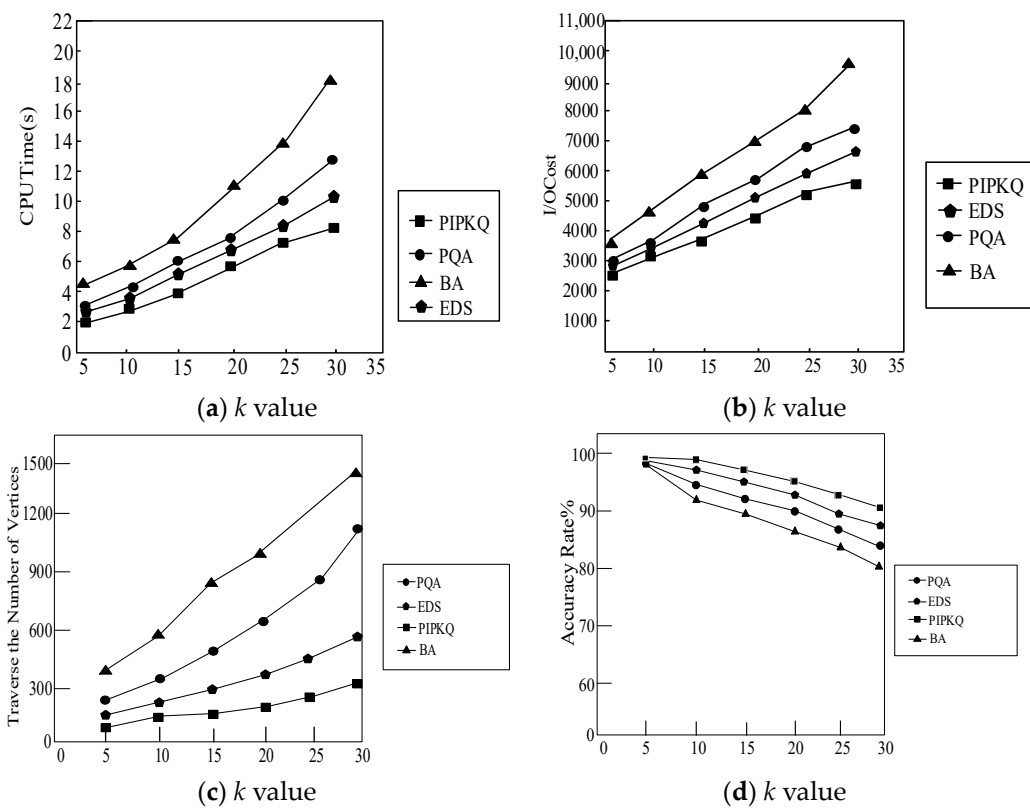

**Figure 10.** The impact of the *k* value on the performance of the query algorithm.

It can be seen from Figure 10a that with the increase of the *k* value, the CPU running time of the PIPKQ algorithm, PQA algorithm, and BA algorithm gradually increases, and the BA algorithm adopts the R-tree index. This algorithm needs to start traversing from the root node, which is the most direct way to solve multiuser preference queries, but the algorithm does not prune the data and other preprocessing operations, and the algorithm takes the most time when the *k* value increases. The PQA algorithm uses an R-tree to build an index, adds additional information to non-leaf nodes, and achieves the purpose of pruning, and the larger the value of *k*, the time the algorithm traverses the index will increase accordingly. The PQA algorithm first clusters multiple users and then dominates and compares the data objects according to the clustering preference order chain and the relaxation granularity. It can filter a large number of data points and adjust the relaxation granularity through the size of the k value, which is generally better than the PQA algorithm and the BA algorithm. The EDS algorithm's influence on algorithm performance in the dimensions of *k* value, its running time also increases with the increase of the *k* value, but the EDS algorithm is superior to the PQA algorithm and BA algorithm. The PIPKQ algorithm in this paper is better than the EDS algorithm. The EDS algorithm also adopts the dominated comparison method, but when the data set size increases, the number of tuples to be compared will increase due to the poor pruning effect. By clustering preference order chain and domination comparison, the PIPKQ algorithm compares the dimensions

in data objects from back to front in order chain order and compares the dimensions that are not related to traditional domination relations so as to prune the data objects with weak domination ability in each dimension and prune a large number of data points that cannot be the result in advance—reducing the number of tuple comparisons. It can be seen from Figure 10b that in terms of I/O cost, the larger the $k$ value, the more times the data object needs to be calculated and compared, and the I/O overhead of the three algorithms gradually increases. However, because the PIPKQ algorithm filters a large number of data points, its I/O cost is better than that of the PQA algorithm, EDS algorithm, and BA algorithm.

As shown in Figure 10c, the number of traversal vertices of the four algorithms increases with the increase of the $k$ value, while the increase of PIPKQ is relatively slow. The PIPKQ algorithm uses the road network Voronoi diagram, and the δ-SNS dominated logarithm data points for filtering and refining. It does not need to traverse every vertex in the road network. Therefore, with the increase of the $k$ value, the traversal of the number of network vertices increases slowly. In terms of query accuracy, this paper defines the ratio of valid query results to all query results as query accuracy. As can be seen from Figure 10d, the accuracy of the four algorithms decreases with the increase of the $k$ value because the larger the $k$ value is, the greater the probability of pruning the data point that may become the result. In the filtering and refining stages, the algorithm proposed in this paper prunes the data that does not tend to the target endpoint of the query planning, so the accuracy is relatively high.

**Experiment 3.** *The impact of the number of data points on query performance. When the number of query objects is 100, the dimension of each query object is 8, and the k value is 5, the PIPKQ algorithm, PQA algorithm, and BA algorithm are compared and analyzed in terms of CPU execution time and I/O cost, as shown in Figure 11:*

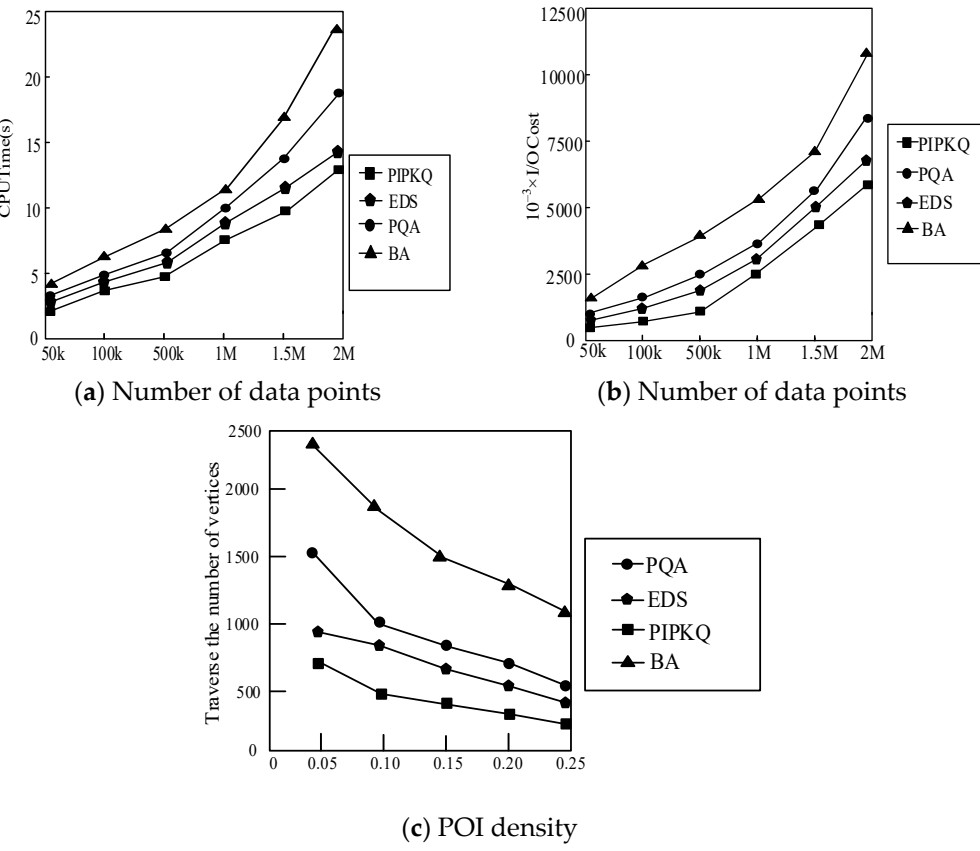

(**a**) Number of data points

(**b**) Number of data points

(**c**) POI density

**Figure 11.** The impact of the number of data points on query performance.

As shown in Figure 11a, as the number of data points continues to increase, the PIPKQ algorithm and EDS algorithm have relatively little running time; the BA algorithm and the PQA take more time. Because the BA algorithm starts traversing data points from the root node of the index, as the number of data points increases, it takes more time to traverse the nodes. The PQA algorithm can prune a small number of data points through an R-tree, but the time and space costs of constructing an R-tree are increasing. The EDS algorithm also adopts the dominated comparison method, but when the data set size increases, the number of tuples to be compared will increase due to the poor pruning effect. The PIPKQ algorithm first clusters the query points and compares the data evolution domination according to the clustering preference order chain; because it adopts a strategy of relaxing granularity, it can filter a large number of non-candidate points, so the CPU running time increases relatively slowly. As shown in Figure 11b, the I/O cost of the three algorithms shows an upward trend, but the PIPKQ algorithm first clusters the query points, then prunes out a large number of data points based on each class cluster, so the rise is slower. The PQA algorithm needs to calculate the query points affected by the same preference, so the I/O overhead is large. The BA algorithm does not preprocess the data points, resulting in high I/O overhead. As shown in Figure 11c, the smaller the POI density, the more vertices traversed by the four algorithms. This is because the smaller the POI density is, the less likely a data point is to be the target point. However, the PIPKQ algorithm traverses fewer vertices than the other three algorithms. The PIPKQ algorithm uses the network Voronoi diagram and δ-SNS to filter and refine the logarithmic data points and directly deletes the data points that cannot be the result, thus reducing the number of vertex extensions.

**Experiment 4.** *The impact of the number of query objects on query performance. When the number of data objects is 10,000, the dimension of interest of each query object is 8, and the k value is 5, the PIPKQ algorithm, PQA algorithm, and BA algorithm are compared and analyzed in terms of CPU execution time and I/O cost, as shown in Figure 12:*

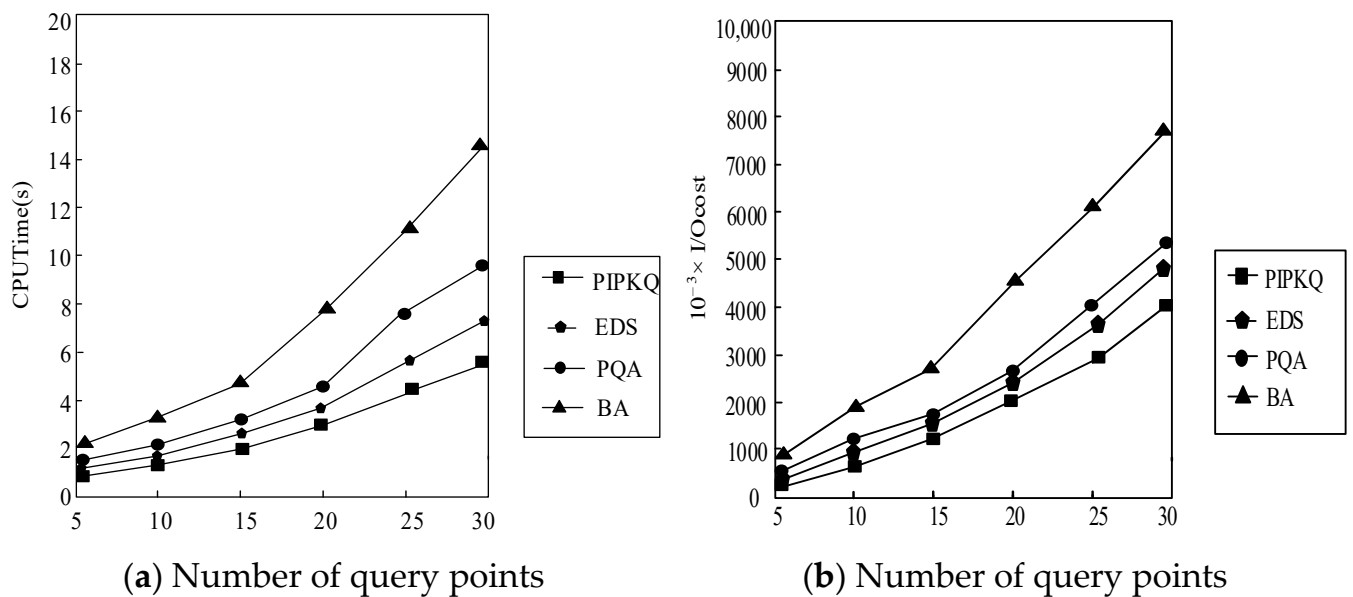

**Figure 12.** The impact of the number of query objects on query performance.

As can be seen from Figure 12a, the running time of the three algorithms gradually increases with the increasing number of query objects. This is because as the number of query objects increases, the corresponding preference increase and more time is needed to process the preference of multiple users. The BA algorithm needs to calculate the possible results of the query objects one by one, and neither the query objects nor the data objects are processed. Therefore, the CPU running time is relatively high. The PQA algorithm

needs to calculate the satisfaction model between each query object and the candidate data object. The EDS algorithm increases relatively slowly, but it is also worse than PIPKQ. It adopts the dominant comparison method, but when the data set size increases, the number of tuples to be compared will increase due to the poor pruning effect. Although the PIPKQ algorithm also consumes time in calculating the similarity between query objects, it filters the data objects according to each cluster's preference order chain, so the run time increases slowly. It can be seen from Figure 12b that with the increase in the number of query objects, the BA algorithm needs to traverse all nodes in the index for each query object, and does not prune the data objects, the I/O cost of the satisfaction model of the PQA algorithm increases, the PIPKQ algorithm groups multiple query objects and then queries the results of each class cluster, avoiding the candidate data objects that traverse each query object, so the I/O overhead increases relatively slowly.

*6.2. Privacy Analysis*

In this section, we use the relative error median (median relative error, MRE) to measure the availability of different methods. In order to verify the availability of our proposed PIPKQ algorithm, this section compares the PIPKQ algorithm with the proportional series allocation method and the arithmetic series allocation method of the privacy budget. The median relative error is:

$$MRE = \frac{P(O) - F(O)}{F(O)} \tag{10}$$

Among them, *P(O)* is the query result obtained by the PIPKQ algorithm that satisfies the differential privacy protection, and *F(O)* is the real query result.

**Experiment 5.** *When the total privacy budget is 0.01, 0.1, 0.5, and 1, and the experimental data set size is 100 K, the relative errors of the PIPKQ algorithm, arithmetic sequence allocation method, and arithmetic sequence allocation method are shown in Figure 13:*

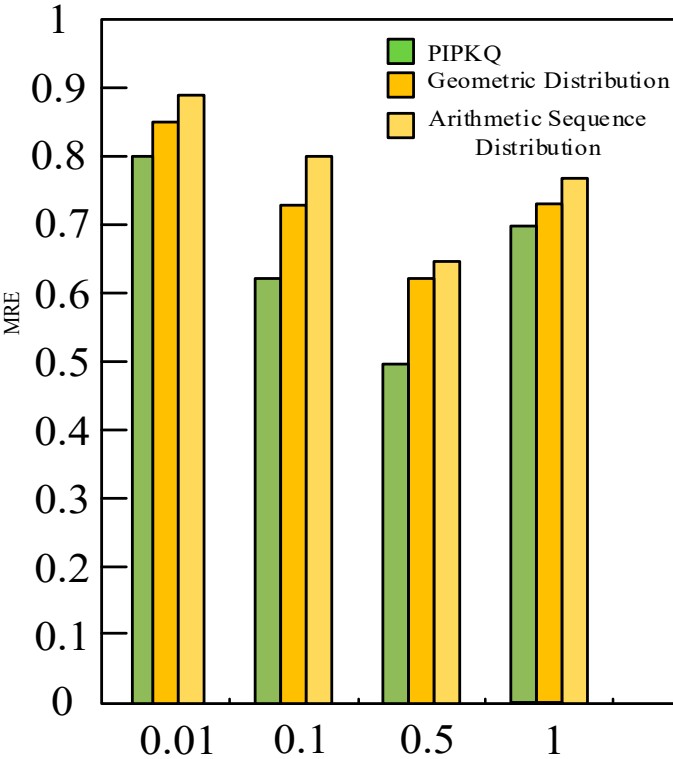

**Figure 13.** Comparative analysis of errors.

It can be seen from Figure 13 that the PIPKQ algorithm is different from the proportional series allocation method and the equal difference series allocation method. The relative error of the PIPKQ algorithm is less than the other two algorithms because the PIPKQ algorithm is allocated according to the weight proportion of each class cluster and will not cause more error. At the same time, the proportional series and the equal difference series allocation method will not consider the characteristics of tuples, which will increase the error. In addition, it can be seen that the relative error between 0.1~1 will be smaller. However, the arithmetic sequence and arithmetic sequence allocation methods only allocate the privacy budget of data according to the characteristics of the method itself without considering the characteristics of the tuple. As a result, the more sensitive data will be allocated, the larger the privacy budget and the lower the degree of privacy protection, thus increasing the error. In terms of privacy protection, how to better balance the privacy and availability of data is the potential problem of this study and also the focus of future research.

**Experiment 6.** *The influence of the k value under privacy protection on the accuracy of data publication. In this experiment, when k changes from 5 to 20, and the remaining variables are default values, After comparing the effects of the algorithm for Cloaking Region [37], the kNN query algorithm for privacy protection through SecureKNNQuery protocol [38] (in this paper, SKQ algorithm for short) and PIPKQ algorithm on the accuracy of result data release under different k values, the experimental results are shown in Figure 14 below.*

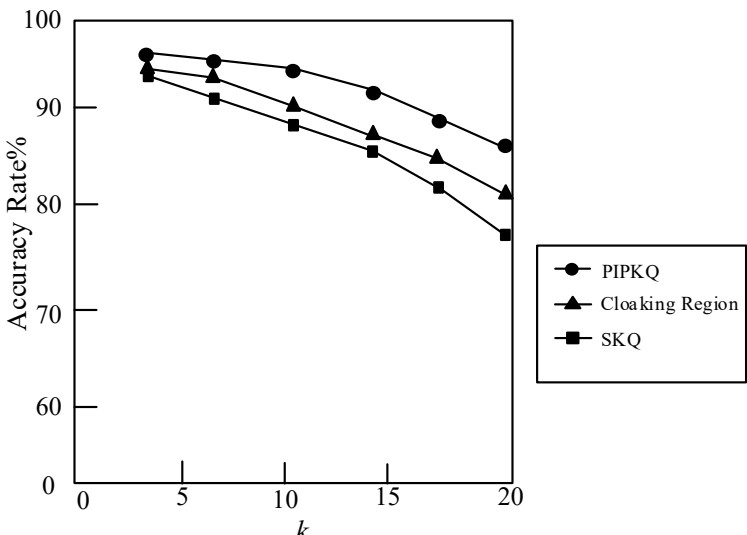

**Figure 14.** The impact of the *k* value on the accuracy of publishing result data.

As can be seen from Figure 14, the query accuracy of the above three algorithms decreases with the increase of the *k* value. However, the accuracy of the PIPKQ algorithm is better than that of the Cloaking Region algorithm and SKQ algorithm. This is because the PIPKQ algorithm uses differential privacy technology to allocate privacy budget by weight ratio, as well as the advantages of differential privacy technology, the published data obtained have relatively high accuracy; The algorithm of Cloaking Region adopts the method of the anonymous box, such that relatively recent data points will not be detected during the query processing, so the query accuracy is relatively low. The SKQ algorithm involves the calculation of the server and the client, and the encryption and decryption complexity of the algorithm is relatively high, and the number of iterations will increase correspondingly with the increase of the *k* value, so the accuracy of the final query results is not high. In summary, the differential privacy protection method has relatively high accuracy with the increase of the *k* value.

## 7. Conclusions

With the development of big data and the advent of the digital age, users' personal preferences are more strongly expressed. Therefore, for query results to meet the needs of different users, mining user preferences in the field of a *k*-nearest neighbor query under the road network environment has become a hot research issue. However, in geographical social network applications, users' personal preferences are potential and uncertain, and it is difficult for users to express them accurately. Moreover, the inaccurate expression of individual users' preferences in multiusers will lead to inaccurate or even incorrect query results. Therefore, this paper proposes a multiusers incomplete preference *k*-nearest neighbor query algorithm based on differential privacy in the road network. First, we obtain the maximal frequent item set according to a HUFP tree. Then, we dig out the positive correlation rules, propose a similarity measurement method according to the positive correlation association rules and road network distance, and cluster the query objects accordingly. Then, the algorithm prunes a large number of data objects based on clustering preference order chain and relaxed granularity, performs a *k* neighbor query on the refined data objects, and finally adds Laplace noise to the query results to protect the data privacy. Experimental results show that the proposed algorithm has good performance and good privacy. Future research work will focus on the following two aspects:

1. Multiuser incomplete preference continuous *k*-nearest neighbor query based on privacy-preserving in road networks.
2. Multiuser incomplete preference *k*-nearest neighbor query based on privacy protection in an edge computing environment.

**Author Contributions:** Conceptualization, Liping Zhang and Xiaojing Zhang; methodology, Liping Zhang and Xiaojing Zhang; investigation, Liping Zhang; writing—original draft preparation, Xiaojing Zhang; writing—review and editing, Xiaojing Zhang, Liping Zhang and Song Li; project administration, Song Li. All authors have read and agreed to the published version of the manuscript.

**Funding:** This research was funded by the National Natural Science Foundation of China (Grant No. 62072136), the Natural Science Foundation of Heilongjiang Province (Grant No. LH2023F031), the National Key R&D Program of China under Grant (Grade No. 2020YFB1710200).

**Data Availability Statement:** The data presented in this study are available on request from the corresponding author.

**Conflicts of Interest:** The authors declare no conflict of interest.

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
