# Peer review of "Multiuser Incomplete Preference K-Nearest Neighbor Query Method Based on Differential Privacy in Road Network"

_ijgi, doi:10.3390/ijgi12070282_

Round 1

Reviewer 1 Report (Previous Reviewer 1)

In this article, the authors propose a k-neighbour query method based on differential privacy based on the incomplete preference of multiple users in the road network. 

I would like to inform you that I had reviewed the initial submission of the article. The authors took into account the reviewers' comments and the article was improved.

The Introduction and the Related Work sections were greatly enhanced.

The contribution of the article has been highlighted, and the steps of the methodology have been captured.

The illustration of the theorems is correct.

The issues with the figures were fixed.

The discussion of the results was enhanced, as well as the conclusions.

The points that need improvement are of minor importance and are as follows.

1)Lines 83-92 concerning the structure of the article should be entered in a separate paragraph.

2)Algorithms should be included in the Appendix section because they create issues with the uniformity of the text. In addition, a table should be created in the appendix with the analysis of the acronyms present in the text.

3)For discussion, I would like to know the limitations and, the potential issues of this work.

Author Response

Point 1: In this article, the authors propose a k-neighbour query method based on differential privacy based on the incomplete preference of multiple users in the road network.

I would like to inform you that I had reviewed the initial submission of the article. The authors took into account the reviewers' comments and the article was improved.

The Introduction and the Related Work sections were greatly enhanced.

The contribution of the article has been highlighted, and the steps of the methodology have been captured.

The illustration of the theorems is correct.

The issues with the figures were fixed.

The discussion of the results was enhanced, as well as the conclusions.

The points that need improvement are of minor importance and are as follows.

  • Lines 83-92 concerning the structure of the article should be entered in a separate paragraph.

Response 1:  Thanks to the expert for your valuable advice, I've made lines 83-92 of the article structure a separate paragraph.

Point 2: Algorithms should be included in the Appendix section because they create issues with the uniformity of the text. In addition, a table should be created in the appendix with the analysis of the acronyms present in the text.

Response 2: Thanks to the expert for your valuable advice, According to the research content of this paper and the template format of this journal, the algorithm of this paper follows the general pattern of this research field. First of all, we put forward a new problem and research method in the field of k-nearest neighbor query, and give the theorem and proof. Then, in each part of the paper, we give the core idea of the algorithm according to the proposed method and theory. Finally, we give the analysis after the algorithm.

Point 3:  For discussion, I would like to know the limitations and, the potential issues of this work.

Response 3: In this paper, multi-user incomplete preference k-nearest neighbor query based on differential privacy in road network, we consider the multi-user preference query in static road network and the preference of multiple query users to the attributes of data objects. Future potential research directions can focus on the multi-user incomplete preference k-nearest neighbor query in dynamic road network and the preference between query users and query users.

Reviewer 2 Report (Previous Reviewer 2)

Authors have delivered a much more complete version, especially in the improved introduction and conclusions.

Author Response

Point 1: Authors have delivered a much more complete version, especially in the improved introduction and conclusions.

Response 1:  Thanks for the expert's approval. I will revise it carefully.

Reviewer 3 Report (New Reviewer)

1. The structure of the overall article writing is complete and the content is rich.

2.The description of the research topic is not very clear. Differential Privacy is mentioned in the title, but the analysis of Differential Privacy is not clearly seen in the article description and experimental results.

3.The K-nearest Neighbor algorithm is a long-ago classification method, and other latest classification algorithms should be added for analysis and comparison.

4.The comparative analysis of the three main algorithms focuses on cpu time and I/O cost, and other related experimental designs are lacking, so it is difficult to see the excellence of the proposed method.

Author Response

Thank you for your suggestion. I have made detailed modifications. Since I cannot upload pictures here, I uploaded the modified document.

Round 2

Reviewer 3 Report (New Reviewer)

The reviewers' comments have been fully responded and the paper has been substantially revised.

This manuscript is a resubmission of an earlier submission. The following is a list of the peer review reports and author responses from that submission.

Round 1

Reviewer 2 Report

In the Introduction, I recommend a mention of the potential implications of the algorithm in terms of speed, accuracy, and privacy for spatial querying applications, as well as typical examples of kNN-querying that could relate with the algorithm.

Line 190 has symbols to edit.

Figure 7 has legend in Chinese.

Reviewer 3 Report

The article is interesting in using frequent itemset and association rules. But the text is not very understandable for readers outside this problem. I suffer with a basic description of the situation and an example of spatial situations in the article. How does it look like a road network?

How look like the user preferences look? Any tables, with the structure? Are they some points pi like in Figure 2?

Please prepare i simple picture or more pictures to explain the problem. Also, the final purpose and utilization of purpose are missing in the article. Is it car navigation or something else? It is somewhere behind the text without explicit saying.

Please put to the introductory part a simple picture with a schematic map of the road network and explain the form of recorded user preferences-how looks like data. Moreover, express how it looks like the incompleteness of user data. And finally, what the query means? What is the object of query from the side of user=driver? Are the queries the points pi?

Maybe simplified Figure 2 put firstly to the Introductory part to explain to readers the starting situation and example data = user preferences - how they look like, any table?

Mistakes:

0) row 5 - What is the proper name of the town Haerbin or Harbin?

1) A lot of spaces between words, brackets etc. are missing in the whole document. The article is full of these errors. The whole article needs very careful checking of text.

2) Carefully check the same version of the term kNN (row 34 - only knn, row 361)

3) References:

- row 43. Why suddenly the number [14] is between [5] and [6]?

- row 104 and from row 128 to the end, the references are from upper letters like [26,27].

4) Chapter 2 Related Work -  There is a very strange use of citations like: Literature [14] put ...

They are very strange citations in sentences introduced several times like this: "Literature [xx] present ....." All they sound like pull-out calling of any facts without logical linking. Please, instead the word "Literature" write the name of the first author.

5) row 92 to 127 is one paragraph. Divide it into logical chunks.

6) Equations (1) add an explanation of D

7) All definitions (rows 170, 176, ...). Remove the brackets from the title of the Definition 1 to the Definition 16 is worse the readability of the text.

8) row 212  ... different, This.... maybe dot as separator

9) Figure 1 and the following description is not understandable. Describe what the left table in the title of the picture is. If Table 1 corresponds to Figure 1, where is p, s, l from Itemsets in the tree?

10) Equations (6) - j is constant? Or is one more Sum for j missing?

11) Figure 2 - Add the legend with an explanation of what means point, and line. What is ri? Prepare it in colour and a bigger to better read the circles.

12) Figure 7 - two items are in the Chinese language. Prepare Figure 7 in colour without hashes - it is very obsolete.